

# Detection and variability analyses of CRISPR-like loci in the *H. pylori* genome

Jerson Alexander García-Zea, Roberto de la Herrán, Francisca Robles Rodríguez, Rafael Navajas-Pérez and Carmelo Ruiz Rejón

Departamento de Genética, Facultad de Ciencias, Universidad de Granada, Granada, Spain

## ABSTRACT

*Helicobacter pylori* is a human pathogenic bacterium with a high genomic plasticity. Although the functional CRISPR-Cas system has not been found in its genome, CRISPR-like loci have been recently identified. In this work, 53 genomes from different geographical areas are analyzed for the search and analysis of variability of this type of structure. We confirm the presence of a locus that was previously described in the VlpC gene in al lgenomes, and we characterize new CRISPR-like loci in other genomic locations. By studying the variability and gene location of these loci, the evolution and the possible roles of these sequences are discussed. Additionally, the usefulness of this type of sequences as a phylogenetic marker has been demonstrated, associating the different strains by geographical area.

## INTRODUCTION

The genus *Helicobacter* comprises 20 formally validated species. Within this group *H. pylori* is particularly important for being a Gram-negative human pathogenic bacterium present in about half of global population (*Gangwer et al., 2010*). This bacterium can induce superficial gastritis and constitutes a risk factor for the development of peptic ulcer disease, gastric adenocarcinome, and gastric mucosa-associated lymphoide tissue lymphoma (*Gangwer et al., 2010*). Due to its characteristics, it can be considered as a model organism for the study of genetics and evolution. *H. pylori* has a great genomic plasticity, presenting high rates of mutation and recombination that allows for the generation of new alleles, allowing it to adapt to relatively specific and well-defined habitats such as the stomach and the duodenum (*Moodley, 2016*). It is well established that *H. pylori* is a highly competent bacterium, and different strains can be found living together in the gastric environment, bringing the populations of *H. pylori* closer to panmixia (*Kang & Blaser, 2006*; *Suerbaum et al., 1998*). Genome comparative analyses from diverse origins have shown that this bacterium shows a high degree of genetic diversity, ranging from nucleotide polymorphisms to genetic mosaicism (*Zawilak-Pawlik & Zakrzewska-Czerwińska, 2017*).

The CRISPR-Cas system is a defense mechanism against foreign genetic elements derived from bacteriophages, plasmids or extracellular chromosomal DNA (*Mojica, Díez-Villaseñor & García-Martínez, 2005*; *Sampson & Weiss, 2013*). The CRISPR-Cas loci are variable in number between bacteria and strains (*Grissa, Vergnaud & Pourcel, 2008*), and its typical

Corresponding author
Jerson Alexander García-Zea,
alexander7719@correo.ugr.es

structure is characterized by a CRISPR matrix, a nearby Cas-gene locus, and an AT-rich leader region (*Zhang & Ye, 2017*). This system is also characterized by its rapid evolution and variability which makes its classification a highly complex task, due to the frequent modular recombination of the CRISPR (*Koonin, Makarova & Zhang, 2017*) matrix, which may mean that not all CRISPR systems carry the same components (*Delaney et al., 2012*; *Grissa, Vergnaud & Pourcel, 2008*) or fulfill the same functions (*Sampson & Weiss, 2013*).

The CRISPR-Cas systems have been identified in approximately 40% of the bacteria and 90% of the archaea. However, *Burstein et al. (2016)* recently proposed that CRISPR-Cas systems are present in only 10% of the archaea and bacteria. This difference in the presence of the CRISPR-Cas system in prokaryotes could due to the fact that the system may not exist in the main non-cultivable bacterial lineages and in those whose lifestyle was symbiotic (*Burstein et al., 2016*; *Burstein et al., 2017*).

In the genus *Helicobacter*, the CRISPR-Cas system has only been detected in *H. cinaedi* and *H. mustelae* (*Kersulyte, Rossi & Berg, 2013*; *Tomida et al., 2017*), both pathogenic species, but not in *H. pylori*. However, *Bangpanwimon et al. (2017)* have more recently described CRISPR-like sequences in the genome of *H. pylori*, more precisely located in the vacA-like paralogue gene (VlpC, HP0922), that could be related to the ability to colonize the stomach (*Foegeding et al., 2016*), suggesting that they could have a regulatory role (*Albert et al., 2005*). In fact, in recent years, hypotheses involving CRISPR loci in the regulation of genes, a function analogous to the functions of RNAi in eukaryotes (*Bondy-Denomy & Davidson, 2014*), have appeared where the CRISPR spacers coincided with genes from the genome itself, with important cellular functions (housekeeping) (*Bondy-Denomy & Davidson, 2014*). Another relationship established between CRISPR and pathogenicity has been discussed in strains of *E. coli* and other species, where the interference of CRISPR prevented the acquisition of virulence genes (*García-Gutiérrez et al., 2015*). On the other hand, a reduced content of CRISPR repeats has also been correlated with a greater likelihood that a strain exerts pathogenicity (potential ability to cause disease) (*García-Gutiérrez et al., 2015*). All of these data exemplify the versatility of CRISPR-Cas systems and suggest roles beyond canonical interference against strange genetic elements (*Hatoum-Aslan & Marraffini, 2014*). The presence of CRISPR orphans of non-vestigial subtype I-F and E in *E. coli* (CRISPR without cas-genes) have been attributed to a possible habitat change, where their presence would be counterproductive (*Almendros et al., 2016*), granting them a regulatory role, whose spacers could prevent the acquisition of cas (anti-cas) genes, thus facilitating the acquisition of genetic material and increasing biological aptitude (*Almendros et al., 2016*).

In this work, we analyze the presence and variability of CRISPR-like sequences in *H. pylori* by studying 53 strains, finding that there are several CRISPR-like sequences in their genomes, which are relatively conserved among strains and can be grouped by geographic area. We discuss their possible role in the generation of variability as well as in the regulation of the genes into which they are inserted.

## MATERIAL AND METHODS

For an analysis of CRISPR-like loci in *Helicobacter pylori*, the sequences of 53 complete genomes (Table 1) (GenBank and fasta formats) of different *H. pylori* strains were downloaded from the genomic resource database of the National Center Biotechnology Information (NCBI) (*Benson et al., 2011*) (ftp://ftp.ncbi.nlm.nih.gov/genomes/). To characterize the CRISPR region in the *H. pylori* genomes we used the CRISPRFinder program with default parameters (*Grissa, Vergnaud & Pourcel, 2007*) (http://crispr.i2bc. paris-saclay.fr/Server/) (Last update on May 9, 2017). In addition, to characterize the CRISPR-like regions in all the genomes analyzed in this work, multiple alignments were created with the Muscle program (*Edgar, 2004*).

We used CRISPRsBlast (*E*-value: 0.01) to determine the similarity between the direct repeats sequences (DRs) and spacers of the CRISPR loci detected in *H. pylori* and the sequences of DRs and confirmed spacers deposited in the BLAST CRISPR database (http://crispr.i2bc.paris-saclay.fr/crispr/BLAST/CRISPRsBlast.php) (*Grissa, Vergnaud & Pourcel, 2008*).

The spacers were also blasted with default parameters against the CRISPRTarget server, which predicts the most likely targets of the CRISPR RNAs (http://bioanalysis.otago.ac.nz/ CRISPRTarget/crispr_analysis.html) (*Biswas et al., 2013*). The databases used were: mobile genetic elements and phages, viruses.

The alignments of CRISPR-like sequences were carried out by Muscle (*Grissa, Vergnaud & Pourcel, 2007*) and Geneious v 6.1.8 (*Kearse et al., 2012*) softwares. The secondary RNA structure and minimum free energy of the DR sequences were predicted using RNAfold WebServer (http://rna.tbi.univie.ac.at/cgi-bin/RNAWebSuite/RNAfold.cgi) with default parameters (*Zuker & Stiegler, 1981*).

For phylogenetic analyses we used the Mega7 program (*Kumar, Stecher & Tamura, 2016*) with the following parameters: Neighbor-joining method with bootstrap of 1,000 replications and Jukes Cantor model.

### Identification of operons linked to CRISPR-like and Cas domains

The research on operons linked to the CRISPR-like structure was carried out using the OperonDB database (*Pertea et al., 2009*) (http://operondb.cbcb.umd.edu/cgi-bin/operondb/operons.cgi).

For the identification of cas domains, the HMMs profiles (Markov Hidden Models Profile) of the Cas families were downloaded from TIGRFAM (ftp://ftp.jcvi.org/pub/data/ TIGRFAMs/) as well as the Cas proteins described by *Haft et al. (2005)*. The search of cas proteins was carried out with HMMER software v3.1b2 (*Eddy, 1998*), implementing the option 'hmmscan' (search in proteins against collections of proteins of the 53 genomes), with an *E*-value 10e−.

### Identification of vacA-like gene (VlpC)

To identify and determine the presence of vacA-like gene (VlpC) in the 53 genomes of *H. pylori* used in this study, the reference sequence of strain J99 (*Chanto et al., 2002*; *Lara-Ramírez et al., 2011*) was downloaded from NCBI: WP_000874591.1 (VlpC). VlpC

García-Zea et al. (2019), *PeerJ*, DOI 10.7717/peerj.6221

Peerj

**Table 1 Characteristics of the CRISPR-like loci detected with CRISPRFinder in the 53 strains of *H. pylori*.** The different colors represent the presence of cas domains in the analyzed genomes : blue: Cas2, red: Cas3, yellow: Cas4 and green: Csa3. Cas1 domains were no tdetected.

| Accesion number | Strain | Origin/isolation | Diagnosis | CRISPRFinder detection | Cas domains detect with Hmmscan (HMMER) *E*-value 10e−5 | | | |
|---|---|---|---|---|---|---|---|---|
| | | | | Gene with CRISPR locus | cas2 | cas3 | cas4 | Csa3 |
| NC_000921 | J99 | Africa/USA | Duodenal ulcer | VlpC | | ■ | ■ | |
| NC_017374 | 2017 | Africa/France | Duodenal ulcer | VlpC | | ■ | ■ | |
| NC_017381 | 2018 | Africa/France | Duodenal ulcer | VlpC | | ■ | ■ | |
| NC_017357 | 908 | Africa/France | Duodenal ulcer | VlpC | | ■ | ■ | |
| NC_017371 | Gambia94/24 | Africa/Gambia | unknown | | | ■ | ■ | |
| NC_017742 | PeCan18 | Africa/Peru | gastric cancer | | ■ | ■ | ■ | |
| NC_017361 | S. africa7 | South Africa | unknown | | ■ | ■ | ■ | |
| NC_022130 | S. africa20 | South Africa | unknown | | ■ | ■ | ■ | |
| NC_017063 | ELS37 | America/El Salvador | Gastric cancer | | | ■ | ■ | |
| NC_017733 | HUP-B14 | Europe/Spain | unknown | | | ■ | ■ | |
| NC_014560 | SJM180 | America/Peru | Gastritis | Hypothetical protein -VlpC- Hypothetical protein | | ■ | ■ | |
| NC_011498 | P12 | Europe/German | Duodenal ulcer | | ■ | ■ | ■ | |
| NC_012973 | B38 | Europe/France | MALT lymphoma | VlpC | | ■ | ■ | |
| NC_011333 | G27 | Europe/Italy | unknown | | | ■ | ■ | |
| NC_021217 | UM037 | Asia/Malasya | unknown | | ■ | ■ | ■ | |
| NC_014256 | B8 | unknown | Gastric ulcer | VlpC | ■ | ■ | ■ | |
| NC_017362 | Lithuania75 | Europe/Lithuania | unknown | | | ■ | ■ | |
| NC_000915 | 26695 | Europe/UK | Gastritis | VlpC | ■ | ■ | ■ | |
| NC_018939 | 26695 | unknown | unknown | VlpC | ■ | ■ | ■ | |
| NC_018937 | Rif1 | Europe/German | unknown | VlpC | ■ | ■ | ■ | |
| NC_018938 | Rif2 | Europe/German | unknown | VlpC | ■ | ■ | ■ | |
| NC_008086 | HPAG1 | Europe/Sweden | Atrophic gastritis | VlpC | | ■ | ■ | |
| NC_022886 | BM012A | Oceania/Australia | Asymptomatic-reinfection | Poly E-rich protein | | ■ | ■ | |
| NC_022911 | BM012S | Oceania/Australia | Asymptomatic-reinfection | | | ■ | ■ | |
| NC_017372 | India7 | Asia/India | Peptic ulcer | VlpC | ■ | ■ | ■ | |
| NC_017376 | SNT49 | Asia/India | Asymptomatic | VlpC | ■ | ■ | ■ | |
| NC_017926 | XZ274 | Asia/China | Gastric cancer | VlpC | ■ | ■ | ■ | |
| NC_020509 | OK310 | Asia/Japan | unknown | | | ■ | ■ | |
| NC_017367 | F57 | Asia/Japan | Duodenal ulcer | VlpC | | ■ | ■ | |

Garcia-Zea et al. (2019), *PeerJ*, DOI 10.7717/peerj.6221

**Table 1** (*continued*)

| Accesion number | Strain | Origin/isolation | Diagnosis | CRISPRFinder detection | Cas domains detect with Hmmscan (HMMER) *E*-value 10e−5 | | | |
|---|---|---|---|---|---|---|---|---|
| | | | | Gene with CRISPR locus | cas2 | cas3 | cas4 | Csa3 |
| NC_017360 | 35A | Asia/Japan | unknown | | | ■ | ■ | |
| NC_017368 | F16 | Asia/Japan | Gastritis | | | ■ | ■ | |
| NC_021218 | UM066 | Asia/Malasya | unknown | | | ■ | ■ | |
| NC_021215 | UM032 | Asia/Malasya | peptic ulcer | | | ■ | ■ | |
| NC_021216 | UM299 | Asia/Malasya | unknown | | | ■ | ■ | |
| NC_021882 | UM298 | Asia/Malasya | unknown | | | ■ | ■ | |
| NC_017365 | F30 | Asia/Japan | Duodenal ulcer | | | ■ | ■ | |
| NC_020508 | OK113 | Asia/Japan | unknown | | ■ | ■ | ■ | |
| NC_017375 | 83 | unknown | unknown | | ■ | ■ | ■ | |
| NC_017382 | 51 | Asia/Korea | Duodenal ulcer | | | ■ | ■ | |
| NC_017366 | F32 | Asia/Japan | Gastric cancer | | ■ | ■ | ■ | |
| NC_017354 | 52 | Asia/Korea | unknown | | ■ | ■ | ■ | |
| NC_014555 | PeCan4 | America/Peru | gastric cancer | | ■ | ■ | ■ | |
| NC_017379 | Puno135 | America/Peru | Gastritis | | ■ | ■ | ■ | |
| NC_017378 | Puno120 | America/Peru | Gastritis | | ■ | ■ | ■ | |
| NC_017359 | Sat464 | America/Peru | unknown | | ■ | ■ | ■ | |
| NC_010698 | Shi470 | America/Peru | Gastritis | Poly E-rich protein | ■ | ■ | ■ | |
| NC_017740 | Shi169 | America/Peru | unknown | | ■ | ■ | ■ | |
| NC_017739 | Shi417 | America/Peru | unknown | Hypothetical protein | ■ | ■ | ■ | |
| NC_017741 | Shi112 | America/Peru | unknown | Hypothetical protein | ■ | ■ | ■ | ■ |
| NC_017358 | Cuz20 | America/Peru | unknown | | ■ | ■ | ■ | ■ |
| NC_017355 | v225d | America/Venezuela | Gastritis | | ■ | ■ | ■ | |
| NC_019563 | Aklavik86 | America/Canada | Gastritis | | ■ | ■ | ■ | |
| NC_019560 | Aklavik117 | America/Canada | Gastritis | | ■ | ■ | ■ | |

gene is located in strain J99 at position 945.691 to 952.890. This sequence was blasted against the 53 *H. pylori* genomes with the following parameters: *E*- value: 10e−, query coverage > 75%.

Also, to determine if the corresponding mRNA of the VlpC gene of the different strains of *H. pylori* was expressed, the cDNA sequences of the 53 genomes were downloaded via FTP (http://bacteria.ensembl.org/info/website/ftp/index.html) and used as a target to be blasted with the CRISPR-like sequences detected in the VlpC gene, using an *E*-value of 10e−.

In addition, for genes that showed a CRISPR-like sequence outside of VlpC, their presence in all genomes was verified using blastn with Geneious v 6.1.8 (*Pertea et al., 2009*) using an *E*-value of 10e−.

To determine genomic rearrangements and possible break-point involved in recombination events, the Mauve software was used for complete alignment of genomes (*Darling, Mau & Perna, 2010*).

## RESULTS

### CRISPR-like loci identification

A total of 53 *H. pylori* assembled and annotated genomes from different geographical regions were analyzed with CRISPRFinder software. Twenty-two CRISPR-like loci were found in 20 strains, with 19 of them exhibiting one CRISPR-like locus and only one strain, SJM180 (Table 1) showing three CRISPR-like loci. Of all loci, 16 were located within a vacA-like gene (VlpC gene), with four DRs and three spacer sequences. This gene was integrated in an operon with the genes *OMP, 4-oxalocrotonate tautomerase, recR, truD, htpX, folE, IspA* and, *surE*, in this order (*Benson et al., 2011*). The remaining six CRISPR-like loci were present in other locations of the genome. More specifically, they were located in: (a) the BM012A (Australian origin) and Shi470 (Peru origin) strains in a Poly E-rich gene rich protein; (b) the Shi417 and Shi112 (both Peru origin) strains within a hypothetical protein (with GO term COG119), and; (c) the SJM180 (Peru origin) strain, with two additional loci, with these located in two different hypothetical protein genes (Table 1). For these 22 loci, which were detected with CRISPRFinder, 95 direct repeat sequences (DRs) were identified, being present in four to seven sequences per CRISPR-like locus and ranging from 23 to 36 bp in length (Table S1). No similarities were found when these sequences were blasted against the CRISPRsBlast database. A total of 73 spacers were detected ranging in number from 3 to 6 sequences per locus, with lengths ranging between 16 to 69 bp. Using CRISPRTarget software, five spacers showed similarities to phage, plasmids or viruses sequences (Table S2).

Consensus DRs for each locus, were generated by CRISPRFinder and sequences for each spacer were used to carry out a phylogenetic study. For DRs, two main groups were observed in the phylogenetic tree: one, including the DRs of the six CRISPR-like loci located out of VlpC gene, and the other with the strains that had the loci within the VplC gene (Fig. 1). The spacers in the phylogenetic tree could be divided into four main groups: three of them corresponded to the group of spacers present in the first, second and third position within

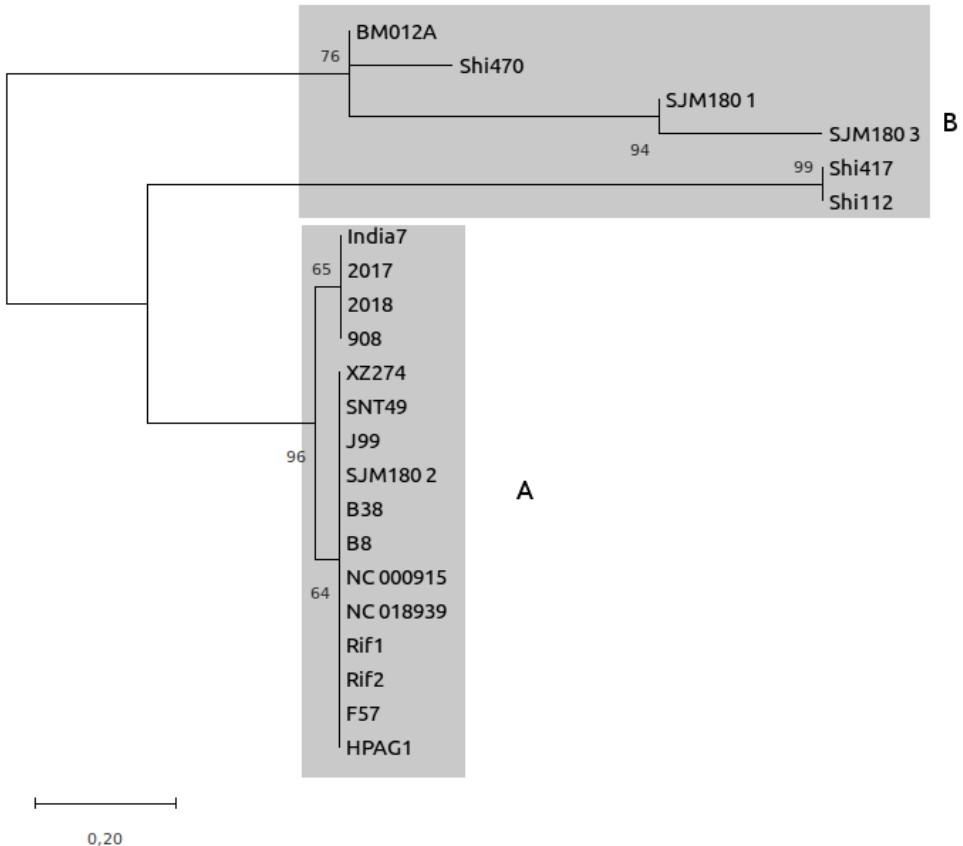

**Figure 1** **Classification of the repeated consensus sequences obtained from CRISPRFinder.** Phylogenetic tree of there peated consensus sequences obtained from the CRISPR loci confirmed to establish evolutionary relationships and classify these sequences. The MEGA7 software was implemented for this analysis. The evolutionary distance scale is 0.2 Jukes-Cantor model. (A) CRISPR located within the VlpC gene. (B) CRISPR located within genes other than theVlpC gene.

the CRISPR-like loci located within the VlpC gene. The fourth group corresponded to the spacers of the CRISPR-like loci found in other genes within the SJM180 (CRISPR1-like and CRISPR3-like), BM012A, Shi417 and Shi112 strains (Fig. 2).

## Analysis of CRISPR-like sequences located within VlpC

The VlpC gene was present in all genomes, except for strain Aklavik86. A manual construction of multiple alignments allowed us to determine the presence of a CRISPR-like structure within the VlpC gene for all genomes. Only the South Africa20 strain showed the VlpC gene but not the CRISPR-like locus, as the gene is truncated in the 5′ region where this structure would be found. The CRISPR-like locus possessed different degrees of variability between strains. The alignment allowed for an in-depth study of DRs and spacers for this locus. It was observed that the variation of the CRISPR-like structure in the VplC gene was mainly due to the complete duplication and/or deletion of spacers and DRs (Fig. S1). The sequences from the 51 CRISPR-like loci detected in VlpC were used to carry out a phylogenetic analysis. Three clusters were observed, created by grouping the

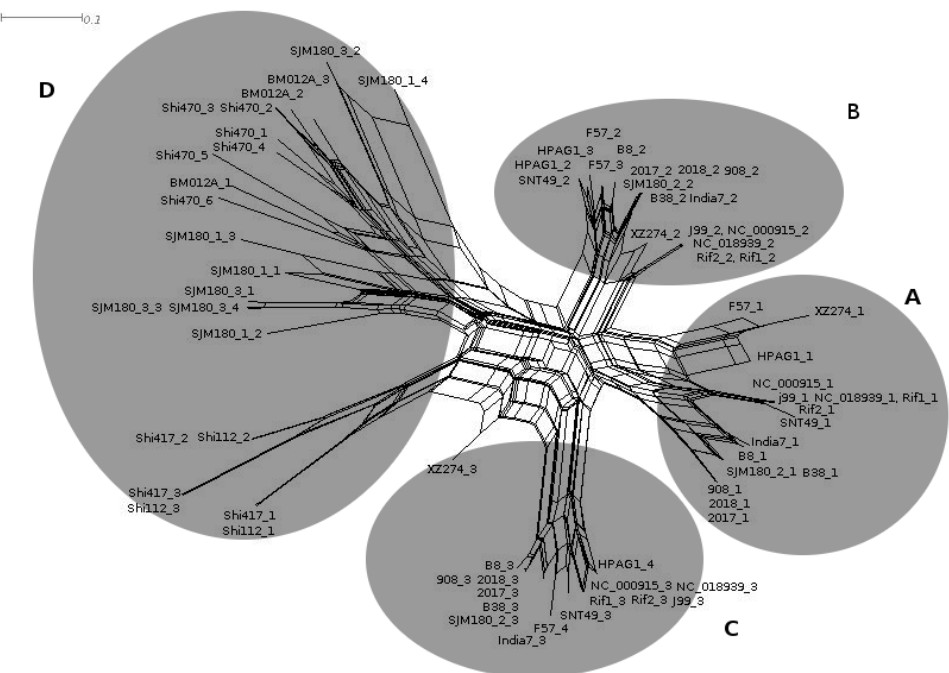

**Figure 2   Classification of the spacers sequences obtained from CRISPRFinder.** Phylogenetic tree for the classification of the spacers sequences obtained from the confirmed CRISPR, based on evolutionary relationships implementing the MEGA7 and software. The evolutionary distance scale is 0.1 Jukes-Cantor model. (A, B, and C) represent the spacers located within the VlpC gene. (D) represents the spacers located within genes other than VlpC.

sequences according to their geographical origins (Fig. 3). The first group included the Africa and Europe strains (group A), the second included the Asia (group B) strains and with the last being the Amerindian strains (group C).

Despite the great variability detected, when the transcriptomes of the *H. pylori* strains were analyzed, it was found that the gene corresponding to VlpC mRNA was expressed in 50 of the 52 genomes that possessed this gene, including the CRISPR-like sequence (Table S3).

When a blastn (*E*-value: 10e−, query coverage > 75%) was performed using the VlpC gene sequence from *H. pylori* against the genomes of other Helicobacter species, only *H. cetorum* showed the presence of this gene. This gene had the CRISPR-like structure, similar to *H. pylori* and an identity above 80% in DRs, indicating that it was the same locus.

## Analysis of CRISPR-like sequences located outside the VlpC gene

In addition to the 16 CRISPR-like loci detected in the VlpC gene by CRISPRFinder, we detected two additional loci in the Shi417 and Shi112 (WP_000536430 and Shi112 WP_000536429 hypothetical protein, respectively) strains, which had identical sequences in their DRs and spacers. These were located in the 5′ region of a gene from a hypothetical protein, between the positions 55,000 to 56,000 of the genome. The ontology analysis showed that this protein had domains related to cell division and cycle control. The

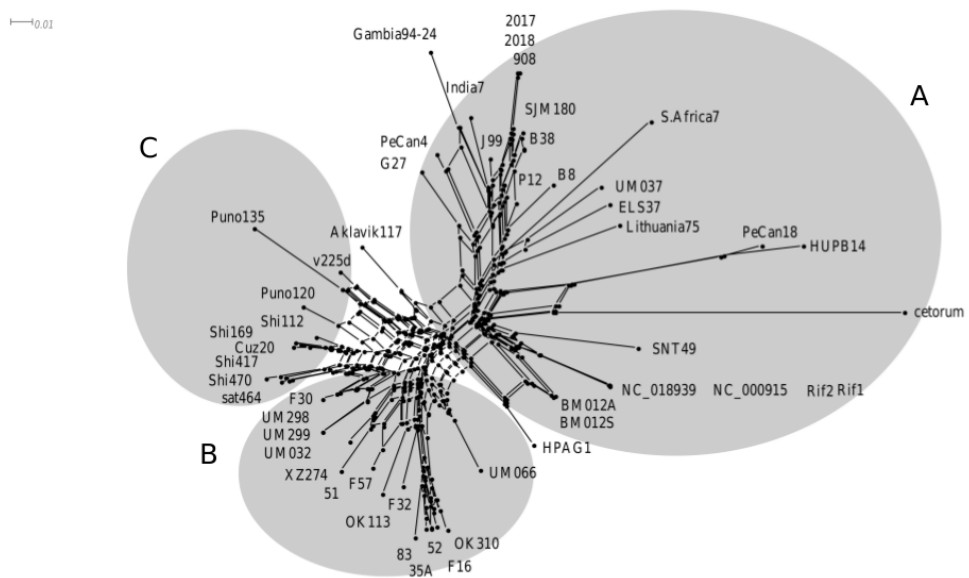

**Figure 3 Classification of CRISPR-like in VlpC gene.** Phylogenetic tree constructed with the 51 CRISPR-like sequences present and located inside the VlpC gene, which evidences a phylogeographic differentiation of the CRISPR-like loci. Analysis executed with MEGA7 software. The evolutionary distance scales is 0.01 Jukes-Cantor model. (A) Group of African and European geographical origin. (B) Geographical group of Asian origin and (C) Amerind geographic group.

CRISPR-like locus of this gene had a length of 150 bp, with four 23 bp DRs and three 19 bp spacers. No similarities were found with other types of genetic element.

When blastn ($E$-value: 10e−, query coverage > 75%) was performed using the sequence of this gene against the remaining 51 genomes, it was found in 12 more strains. The 5′ regions were low conserved, and even three strains (aklavik86, aklavik117 and P12) had this region truncated. Whereas the 3′ region was highly conserved (85%) for the twelve strains (Fig. S2). All the genes had a CRISPR-like locus in their sequence but, as the CRISPR-like locus is located in the 5′ region of the genes, they were degenerate (56% of identity). The origins of these 14 strains with CRISPR-like locus were Amerindian (6) European (6) and African (2), and the phylogenetic tree constructed, using the CRISPR-like sequences, clearly separated these three groups (Fig. S3).

The Shi470 and BM012A strains showed a CRISPR-like locus within a Poly-E rich protein gene (WP_00078209, WP_023591955 respectively). In Shi470, this gene was located between the positions 320.726 and 322.187. In the case of BM012A, it was between the positions 659.636 and 661.240. In a genomic structural analysis of these two strains with Mauve software (*Darling, Mau & Perna, 2010*), it was verified that it was the same gene present in a syntenic region but affected by a genomic rearrangement. This gene was included in an inverted segment and near a breakpoint (Fig. S4A). The alignment of this gene from both strains revealed a middle location of the CRISPR-like locus, with a 70% similarity. The divergences could be explained for the different number of DRs and spacers detected among them (Fig. S4B). The CRISPR-like locus of Shi470 had a length of 660 bp

with seven DRs and six spacers while that for BM012A was 174 bp in length with four DRs and three spacers. It was interesting to note that the spacers of the Shi470 strain showed similarity with mobile elements and phages (Table S2), while no similarities were found for those from the BM012A strain.

A blastn (previous parameters) search with the rest of the genomes (51) allowed us to identify this gene in 33 more strains, all of them with CRISPR-like features. The genes showed a high identity (close to 80%) in their sequence except in the CRISPR-like region (60% identity) (Fig. S5). The phylogenetic tree, constructed with the CRISPR-like sequences, clearly separates the four geographic regions (Fig. S6).

In relation with the two additional CRISPR loci detected by CRISPRFinder in SJM180 strain, these were called CRISPR1-like and CRISPR3-like and were found in two different hypothetical proteins (WP_000446591-CRISPR1-like; WP_013356447-CRISPR3-like). Their percentage of identity was not significant for considering that they were the same gene. The CRISPR1-like loci was inserted in the middle of the gene and was located in position 128.894 to 128.614 of this strain's genome, with a length of 314 bp, five DRs (with an average length of 25 bp), and four spacers (with a length of 34 bp). Spacers 1_1 and 3_1 showed similarity with plasmids and viruses, respectively (Table S2). A blastn search for this gene revealed that this protein is present in 39 strains. Also, it was observed that the sequence from region 5′ to the beginning of CRISPR1-like (approximately 380 bp) was highly conserved (91%), while the region corresponding to CRISPR-like was degenerate (63%), with the 3′ region (approximately 600 bp) being highly conserved (90%) as well (Fig. S7). The phylogenetic tree using the 39 CRISPR1-like sequences showed, in this case, a mixture of the strains in relation to their geographical origin (Fig. S8).

The CRISPR3-like region, with a length of 266 bp, was also inserted in the middle of the gene (positions 1.201.946 to 1.201.720 of the genome) and showed five DRs (average length of 23 bp) and four spacers (ranging between 19 and 31 bp), with spacers 1 and 3 showing similarity with plasmids (Table S2). The blastn analysis revealed this protein to be in 27 strains. This hypothetical protein was highly conserved (96%) from the 5′ region to the beginning of the CRISPR-like region (approximately 380 bp) while the CRISPR3-like region was degenerate (61%) and the 3′ region (approximately 520 bp) was conserved (81%) (Fig. S9). The phylogenetic tree created with these 27 sequences showed, as in the previous case, a mixture of the strains of different geographical origins (Fig. S10).

Additionally, the DRs consensus from all CRISPR-like loci found in *H. pylori* were analysed using RNAfold Server (*Zuker & Stiegler, 1981*), and the secondary RNA structure predicted. Also, the minimum free energy was found to range from −0.74 to −7,73 kcal/mol (Fig. S11).

## Cas Domain detection

Cas3 and Cas4 domains were identified in 100% of the analyzed strains, whereas Cas2 domains were found in 32 strains (60.4%), and the Csa3 domain only in two strains (4%) (Table 1, Table S4). These domains were found in various locations in the different strains.

## DISCUSSION

In the genome of the human pathogenic bacterium Gram negative, *H. pylori*, the CRISPR-Cas system is not functional and does not exist by forming an operon structure as it is known for other organisms. The lack of this system in some prokaryotes has been related to the increase in the capacity to integrate exogenous DNA in the genome of these bacteria and, resulting in the acquisition of new functions, which can confer an adaptive advantage to these strains, particularly during their transition to pathogenesis (*Sampson & Weiss, 2013*; *Sampson & Weiss, 2014*). But recently, *Bangpanwimon et al. (2017)* reported the presence of CRISPR-like sequences inserted into the VlpC gene of *H pylori*. In that study, the detection was performed by PCR in partial regions of the genome of Thailand isolates (*Bangpanwimon et al., 2017*). Each isolated strain showed a CRISPR-like locus with similar DRs sequences. However, results from other strains from different geographical regions, the variability of this locus, or the possibility of the presence of other CRISPR-like loci in the genome of *H. pylori* were not analyzed.

In this work, we show the analysis of 53 strains of *H. pylori* which comprise all the continents. The phylogenetic analyses carried out using the sequences of the CRISPR-like locus found by CRISPRFinder revealed the existence of additional loci to the CRISPR-like locus inserted into the VlpC gene described by *Bangpanwimon et al. (2017)* (Figs. S2, S5, S7, and S9).

Of the 53 genomes analyzed, 51 of them showed a locus similar to the CRISPR-like locus found in VlpC gene, with DRs and spacer sequences similar to those detected in *Bangpanwimon et al. (2017)*. In the phylogenetic tree, using the CRISPR-like sequences present in this gene, we observed that the strains that corresponded to an African and European origin formed a differentiated cluster with respect to the Asian and Amerindian strains (Fig. 3). This fact would indicate that the strains furthest from the African origin, such as those of Asian and Amerindian origin, have undergone a process of greater differentiation. *Duncan et al. (2013)* proposed that the different strains of *H. pylori* were subject to different selective pressures depending on their environmental conditions and according to their phylogeographic origin, and this can lead to the diversification of certain genomic regions, as seems to be the case here (*Kang & Blaser, 2006*).

The presence of CRISPR-like loci caused changes in the sequence of the genes where they are inserted into, truncating it or varying its sequence close to the insertion point (Figs. S2, S5, S7, and S9). In addition, the CRISPR loci themselves showed great variability between strains because DRs and spacers were variable in number, even with reverse positions in several genomes (Figs. S2, S5, S7, and S9). These variations indicate recombination phenomena that involve the CRISPR-like locus. In this sense, the CRISPR-like loci could be considered as repetitive sequences involved in intra- and inter- genomic recombination, contributing to the diversity of *H. pylori*. In fact, the variability found between strains, with duplications and deletions within DRs and spacers, could be the result of both types of recombination. In addition, in this work we showed the presence of a CRISPR-like locus in a region near the breaking point of a large inversion that affects several strains (Shi470 and BM012A), and may therefore be involved in this process (Fig. S4).

In *Helicobacter* there have been reports about the implications of repeated sequences in this type of rearrangement events (*Kang & Blaser, 2006*; *Aras et al., 2003*; *Suerbaum & Josenhans, 2007*). The implication of CRISPR-like loci in the recombination process could also be supported by the presence of a RecR gene, which is implicated in recombination and repair processes (*Marsin et al., 2008*), in the same operon as the VlpC gene.

All of these processes would be part of the mechanisms that infer the extreme genome plasticity of *H. pylori* through mutation and recombination intra e inter genomic, exhibiting genetic mosaicism (*Suerbaum et al., 1998*).

Currently, it is hypothesized that degenerated CRISPR-Cas systems, or their individual components, as in this case, could derive into diverse roles in a wide range of processes (*Mojica, Díez-Villaseñor & García-Martínez, 2005*). Thus, if a novel function of a CRISPR system, or one of its components, confers a competitive advantage in the environment in which the organism evolved (that is, it is adaptive) its maintenance and propagation in populations could be a direct result of natural selection. The analysis of secondary structure of direct repeat RNA showed that all sites can form stable RNA secondary structure, exhibiting stem-loop structures and low minimum free energies. The presence of these structures would suggest a possible role in recognition-mediated contact between gap targeted RNA, DNA or protein (*Zhao, Yu & Xu, 2018*).

In fact, it has been shown that orphan CRISPRs loci may be involved in gene regulation. In *Listeria monocytogenes*, orphan CRISPR affected virulence through the FeoAB iron transport system (*Mandin et al., 2007*). In this sense, the constant presence of this repeated and mutable structure in these genes of *H. pylori*, and more specifically in the VlpC gene, which is part of the central genome, could be related to the regulation of its expression, as they are located in the promoter region. The integration of the CRISPR-like structure into the VlpC gene would allow the bacteria to be less sensitive to the host defense mechanisms as indicated by *Bangpanwimon et al. (2017)*, and would confer the ability to adapt to different stomach areas, facilitating the capacity to adhere to the gastric epithelium (*Harvey et al., 2014*). Similar situations have been described in *Staphylococcus aureus*, in which case the absence of the CRISPR-Cas system conferred the ability to acquire new genes and be more virulent, or as *Enterococus fecalis*, where the modification of their CRISPR-Cas systems made their strains more resistant to antibiotics (*Sampson & Weiss, 2013*)

Although clustered Cas genes were not detected, and therefore a functional CRISPR-Cas system was also not found, in this work the presence of cas domains in the genome of *H. pylori* was found (Table S4). This presence could signify that the presence of this system is ancestral. This theory could be strengthened by the fact that in other Helicobacter species, CRISPR-Cas systems are present and active (*Kersulyte, Rossi & Berg, 2013*; *Tomida et al., 2017*). The cas domains in the *H. pylori* genome could be performing other functions. In fact, it has been reported, in *H. pylori*, that the VapD protein, associated with a ribonuclease function, is phylogenetically related to Cas2 proteins. Specifically, the HP0315 protein, a member of the VapD family, has a structural similarity to Cas2 and appears to be an evolutionary intermediate between Cas2 and a gene from the Toxin-Antitoxin system (*Kwon et al., 2012*).

The loss of the functional system is also supported by the fact that in the evolutionary process the number of repetitions present in a CRISPR locus depends on the level of decay of the associated genes (*Touchon & Rocha, 2010*), as is the case of *H. pylori*, in which the number of DRs observed is low and only cas domains are found, which may be remnants of the original system.

From the analysis carried out, the presence of a CRISPR-like locus within several genes of *H. pylori* was demonstrated. The origin and evolution of these types of sequences is still uncertain. However, for the case of the structure found in the VlpC gene, data is available that has helped with inferring its evolutionary history. In this sense, when comparing the genomes of different Helicobacter species, it was found that the VlpC gene was only found in *H. pylori* and *H. cetorum* and with a high degree of similarity. This could indicate that this gene was acquired after the separation of the common ancestor of *H. pylori* and *H. cetorum* from the rest of the species, by duplication from the vacA gene (*Foegeding et al., 2016*). After this event, the acquisition of the CRISPR-like sequences could have taken place in the VlpC gene. These structures, as pointed out, are in a state of constant flow (*Marraffini, 2013*), and therefore they can appear and disappear depending on the selective forces of the environment. During the speciation process of *H. pylori* and *H. cetorum*, the differentiation of CRISPR-like loci occurred between both species. In this sense, it could be said that although the DRs of both species have a high degree of similarity, indicating the common origin, the spacer sequences are variable. It has also been suggested that CRISPR loci can evolve rapidly in some environments, in accordance with the new role played in their antagonistic coevolution (*Westra et al., 2016*).

The CRISPR-like loci in *H. pylori* have evolved independently of those of *H. cetorum* (sympatrically), supporting this type of antagonistic coevolution.

In addition, and due to the high degree of change found in these sequences (Fig. S1), CRISPR-like loci can be used to determine a strain's origin. Different genomic regions have been used for phylogenetic analyses of *H. pylori* as Multi Locus Sequence Typing (MLST), *housekeeping genes and* genes of the central genome (*Falush, Stephens & Pritchard, 2003*; *Falush et al., 2003*; *Yahara et al., 2013*). In our case, the phylogeny, using DRs and spacers of CRISPR-like locus within the VlpC gene, groups the strains by geographic origin (Fig. 3), relating the African ones with the European ones, separating them from the Asian and Amerindian ones (of more recent origin). This same situation was observed for the CRISPR-like loci of the Poly-E rich poly genes and for one of the hypothetical proteins (with cell division function), with a grouping by geographical origin (Figs. S3 and S6). For this latter protein, the absence of this gene in all strains of the Asian clade was highlighted. Lastly, the sequences of CRISPR1-like and CRISPR3-like loci did not have a geographical grouping (Figs. S8 and S10), showing a process of variation that was independent of the geographical origin.

## CONCLUSIONS

We detected the presence of different CRISPR-like loci in almost all the analyzed genomes of *Helicobacter pylori* strains with different geographical origins. We characterized their

structure as well as their location within the genome. The presence of this type of CRISPR-like causes modifications in the genes where they have been inserted, through duplications and deletions of DRs and spacers, loss of 3′ region of the gene, or nucleotide variation caused by deletions, insertions, and substitutions. These mutations increase the variability and in some cases produce genomic rearrangements. Additional studies of these loci, including a high number of strains and its association to phenotypes or other molecular markers, will facilitate bacterial typing. In fact, in this study variations of these loci have been associated with geographical regions. Although the function of this type of loci is unknown, several roles have been proposed for this type of structures. For all this, this work suggests a possible role of this type of sequences, which seem to have lost their initial function, in the variability and genomic evolution of *H. pylori*.

### Funding
The authors received no funding for this work.

### Competing Interests
The authors declare there are no competing interests.

### Author Contributions
- Jerson Alexander García-Zea conceived and designed the experiments, performed the experiments, analyzed the data, contributed reagents/materials/analysis tools, prepared figures and/or tables, authored or reviewed drafts of the paper, approved the final draft.
- Roberto de la Herrán and Carmelo Ruiz Rejón conceived and designed the experiments, analyzed the data, contributed reagents/materials/analysis tools, prepared figures and/or tables, authored or reviewed drafts of the paper, approved the final draft.
- Francisca Robles Rodríguez and Rafael Navajas-Pérez prepared figures and/or tables, authored or reviewed drafts of the paper, approved the final draft.

### Data Availability
 The genome sequences of the *H. pylori* and non-pylori strains [J99, 2017, 2018, 908, Gambia94, PeCan18, South Africa7, South Africa20, ELS37, HUP-B14, SJM180, P12, B38, G27, UM037, B8, Lithuania75, 26695 (NC_000915), 26695 (NC_018939), Rif1, Rif2, HPAG1, BM012A, BM012S, India7, SNT49, XZ274, OK310, F57, 35A, F16, UM066, UM032, UM299, UM298, F30, OK113, 83, 51, F32, 52, PeCan4, Puno135, Puno120, Sat464, Shi470, Shi169, Shi417, Shi112, Cuz20, v225d, Aklavik86, Aklavik117 and Helicobacter cetorum Mit 99-5656] are available in the public database (NCBI). Accession numbers can be found in Table 1.

### Supplemental Information
Supplemental information for this article can be found online at http://dx.doi.org/10.7717/peerj.6221#supplemental-information.

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
