# Peer review of "Detection and variability analyses of CRISPR-like loci in the H. pylori genome"

_PeerJ, doi:10.7717/peerj.6221_

## Round 0.1 · original submission · Minor Revisions

· Academic Editor

Minor Revisions

As you will see, your manuscript has been reviewed by 2 reviewers and they have both asked for more details on the methodologies and analyses used. Please address all of their questions in your revised manuscript.

·

Basic reporting

1. In general, the language used in the article is clear and proper. There are point sections that require attention, such as:
a. Line 99: seems to be missing an article.
b. Line 101: Correct National Biotechnology Information Center for National Center for Biotechnology Information (it is after all NCBI, not NBIC).
c. Line 199: an unnecessary “of” at the end of the sentence.
d. Line 253: Correct Csa3 for Cas3.
e. Lines 163, 178: Correct VplC for VlpC.
f. Lines 127, 146: Standardize the notation for VacA-Like gene. In line 127 appear non-capitalized.
g. Lines 136, 187, 201, 222, 234, 244: In the first three, the local alignment nucleotide algorithm is presented as “blastn”. In the other three it is capitalized. The former is the customary (and original) version.
h. Various lines: Italicize multiple occurrences of “Helicobacter” or “H. pylori” that appear in regular text (I stopped keeping tabs after the first few).

2. The range of subjects presented by the cited references is adequate for the thematic content of the article.
a. Standardize the formats in the list of references. Some journal names appear in italics, others in regular text. PLoS is presented in reference 29 as PloS.
b. Reference 5 is missing the journal name and volume number.

3. Professional article structure, figs, tables. Raw data shared.
a. The range of figures is adequate for what the authors hope to illustrate. The images are of low-resolution, making it hard at times to discern the details (e.g., Fig3 or FigS1b).
b. The legend to Figure 3 cites three groups, which one assumes are the three gray bubbles that group the branches of the phylogenetic tree presented, but the groups are not labelled to indicate their correspondence.
c. Supplemental figures need legends. The details contained within each one of them are not sufficiently clear when cited in the main text, so more clarity is needed in a legend.
d. All raw data is publically available. The authors thus fulfill this requirement.
4. Self-contained with relevant results or hypotheses.
a. The article represents a coherent and complete (within the scope of the design) unit of work. The design is consistent with the hypotheses presented, and the results speak to the questions posed.

Experimental design

1. The article presented constitutes original and primary research within the scope of the journal.
2. The research question is well defined, originating from an identified gap in the knowledge of the organism in question, and anchored in the recent literature. The work identifies the CRISPR-like elements in the genomes of a broadly distributed sample of Helicobacter pylori strains, a species in which these elements had been reported as not functional and thus perhap non-present. With the identification of the vestigial elements, they expand provide additional information for the reconstruction of the role of the vestigial elements in the process of genome evolution. The authors place the findings in an evolutionary context, noting biogeographical associations and phylogenetic associations behind the patterns observed.

On line 114, regarding the construction of phylogenetic trees, one is to assume that all other parameters not explicitly mentioned were used with default values, right? Also, the evolutionary model is indicated, but not the type of tree that was constructed. Please provide more details to enable replication of your results.

On the re-construction of phylogenetic relationships among spacers (results reported starting on line 160), it is not clear how the "consensus" sequences for the spacers in group 4 were constructed. Which of the various spacers were used? Were all of the spacers in the non-VlpC locus considered, or also only the first three positions? Please clarify the approach for the reconstruction of that phylogeny.

Validity of the findings

The data are robust, originating from stardized public repositories. The derivations therefrom are in accordance with questions addressed, following standard methods and tools in bioinformatic sequence comparison and analysis. It is not clear, however, that the divergence patterns observed in the CRISPR-like elements represent a new tool for resolving standing biogeographical questions, or simply re-inforce the patterns previously observed, so caution must be exercised to not overinterpret the implications of the observations. As a closing statement the authos declase "this work highlights the importance of this type of sequences, which seem to have lost their initial function, in the variability and genomic evolution of H. pylori". Care must be used with the word "importance" as it relates to the roles suggested, for while the elements may be playing an incidental (though not necessarily inconsequential) role in the genomic evolution of H. pylori, and even if they have been suggested as direct participants in regulatory functions by other authors, their "importance" in the biology of the organism is not directly addressed by the observations presented.

·

Basic reporting

clear

Experimental design

clear

Validity of the findings

clear

Additional comments

Review of “Detection and variability analyses of CRISPR-like loci in the H. pylori genome”
Full paper summary:
This article uses Helicobacter pylori, one of the important human pathogens, to collect the entire genome of 53 strains of H. pylori from different regions. Although the functional CRISPR-Cas system has not been found in the genomes of these strains, CRISPR like loci have been recently identified. Based on this, the authors performed bioinformatics analysis on the CRISPR-like loci present in these strains. In this work, the authors analyzed 53 genomes from different geographic regions for searching and analyzing the variability of this CRISPR-like locus structure. This article confirms the existence of a CRISPR-like locus present in the VlpC gene in all genomes previously reported, and the authors also characterize new CRISPR-like loci in other genomic locations. In addition, the evolution and possible role of these loci were discussed by studying the variability and gene location of these CRISPR-like loci. Finally, the authors confirmed the feasibility of the CRISPR-like locus sequence as a phylogenetic marker for H. pylori and associated different strains with geographic regions.

Review opinion: (Final decision: substantial revision)

1. There are only 53 species of Helicobacter pylori currently found. Why not choose all kinds of Helicobacter pylori for bioinformatics analysis to improve the persuasiveness of this conclusion?
2. Authors must indicate when to collect relevant CRISPR information on the CRISPRs Database.
3. The authors should describe in more detail the virulence, virulence, and effects on human health of Helicobacter pylori.
4. In P114-115, the authors need to explain in more detail how to build a phylogenetic tree, such as distance-based methods using Minimum Evolution or Neighbor-Joining. In addition, using the nucleic acid sequence to construct a phylogenetic tree, the Kimura 2-parameter model is generally used. Why should the author choose the Jukes Cantor model?
5. Please indicate whether there is a palindrome in the repeat sequence in the selected CRISPR locus?If it exists, it is necessary to discuss the RNA secondary structure that it can form.
6. In P128, the author must point out the basis for choosing the reference sequence of strain J99?
7. In P234-240, the author must clearly explain the 39 CRISPR1-like sequences mentioned in this, is this contrary to only the 22 CRISPR-like loci found in P144-145?
8. The author must explain what principles are used to distinguish between CRISPR sites and CRISPR-like loci when analyzing the genomes of these strains.
9. In P309-311, the authors are asked to elaborate on the potential of CRISPR-like loci to regulate gene expression.
10. Please add a description about the diversity of CRISPR-like loci caused by genetic degradation, genetic recombination, gene insertion, etc., and its application in bacterial typing.
11. Please increase the resolution of figure 3, Supplementary_Fig_S1_A, Supplementary_Fig_S1_B to enhance readability.
12. The related publication on CRISPR Loci of foodborne pathogens below should be cited to discuss the research background and results.
Study the Features of 57 Confirmed CRISPR Loci in 38 Strains of Staphylococcus aureus. Front. Microbiol. 9:1591. doi: 10.3389/fmicb.2018.01591

---

## Round 0.2 · accepted · Accept

· Academic Editor

Accept

Thank you for addressing all of the reviewer's comments. I look forward to seeing your work online.

#